# The German Climate Verdict, Human Rights, Paris Target, and EU Climate Law

**Felix Ekardt** [1,2,*] and **Marie Bärenwaldt** [1]

1   Research Unit Sustainability and Climate Policy, 04229 Leipzig, Germany;
    mail@sustainability-justice-climate.eu
2   Faculty of Law and Interdisciplinary Faculty, Rostock University, 18051 Rostock, Germany
*   Correspondence: felix.ekardt@uni-rostock.de; Tel.: +49-341-49277866

**Abstract:** The German Constitutional Court's climate verdict provided a re-interpretation of core liberal-democratic concepts, and it is highly relevant for liberal constitutional law in general, including EU and international law—where similar issues are currently being discussed in ongoing trials before the European Court of Human Rights and the International Court of Justice. The present article applies a legal interpretation to analyse the national and transnational implications of the ruling. The results show that the verdict accepts human rights as intertemporal and globally applicable. It applies the precautionary principle to these rights and frees them from the misleading causality debate. However, the court failed to address the most important violations of human rights, it categorised climate policy as a greater threat to freedom than climate change, and the court failed to acknowledge that the Paris 1.5-degree limit implies a radically smaller carbon budget. Furthermore, little attention has so far been paid to the fact that the ruling implies an obligation for greater EU climate protection, especially since most emissions are regulated supranationally. Against this backdrop, the EU emissions trading system demands a reform, which has to go well beyond the existing EU proposals so as to enable societal transformations towards sustainability.

**Keywords:** climate change; Paris Agreement; human rights; IPCC; climate policy; climate litigation; precautionary principle; climate justice





## 1. Introduction

Since political majorities have only to a limited extent adopted policies that are in line with the 1.5 degree target from Article 2(1) of the Paris Agreement [1,2], an increasing number of supreme courts discuss the potential obligations of political majorities regarding the climate catastrophe (for cases worldwide see the Columbia Law School database, Sabin Centre for Climate Change Law, http://climatecasechart.com, accessed on 6 June 2023) [3,4]. The climate verdict of the German Federal Constitutional Court (FCC) of 24 March 2021, published on 29 April 2021, is a German example [5] (for an analysis, see [6–12]). The ruling is arguably (one of) the most far-reaching ruling(s) on climate protection worldwide that has ever been passed by a supreme court. In any case, the public perception of the ruling took place on a global scale.

The FCC verdict 'Göppel et al.' was issued in 2021 in response to four constitutional complaints [13–16]. The first constitutional complaint was initiated and funded by the Solar Energy Support Association Germany (SFV) in 2018, and it was filed with individual complainants, such as the former Christian Democrat member of the German Bundestag Josef Göppel and Friends of the Earth Germany. The complaint was legally represented by Felix Ekardt and Franziska Heß, and it was prepared on the basis of multiple legal opinions that were written by Felix Ekardt since 2010 (see [17]). These legal opinions are based on the postdoctoral thesis of Felix Ekardt 2019 ([18]; see the shortened and updated version in English in [19]), which has been further developed in several editions and is dedicated, among other things, to the relationship between freedom and sustainability (and

especially climate protection). For a long time, the idea of such a constitutional complaint was not taken seriously by the public, nor by politicians and legal experts—that is until the constitutional complaint was accepted for decision by the FCC in August 2019. Further constitutional complaints were submitted in January 2020 (following the name of one of the complainants who joined only in 2020, some—in a slightly misleading way—call the FCC verdict "Neubauer et al."). All of the constitutional complaints argue that Germany has to improve climate protection. In particular, the complaints demand more ambitious targets than the one established in the German Klimaschutzgesetz (KSG/climate protection act). The law requires minus 55 percent emission reductions by 2030 compared to 1990. But Germany has already achieved around 15 percent emission reduction through the German reunification in the 1990s and via the collapse of the East German industry. The remaining emission reductions are attributed to the emission shifts from Germany to developing countries (see, in detail [18–20]). Against this background, this article analyses the statements, justifications, weaknesses, as well as the political—especially European—consequences of the FCC ruling.

The German Constitutional Court's climate verdict is highly relevant for the liberal constitutional law of other nation states, as well as EU and international law, as it provides a new interpretation of liberal-democratic core concepts. At the centre stand freedom in general—including its intertemporal and global dimensions—, defensive and protective freedom, the precautionary principle, separation of powers, legislatory balancing limits, and rules for dealing with uncertain facts (we will see below that the verdict has an indirect effect throughout the EU and ultimately beyond). Furthermore, the ruling discusses core issues of the 1.5-degree Celsius limit that is based on Article 2(1) of the Paris Agreement which is the guiding star of global climate policy. Our contribution analyses insights, political consequences, and drawbacks of the verdict. These aspects are relevant for a multitude of academic disciplines in the human sciences and for society on a transnational level—they are by no means only for lawyers.

Our article is structured as follows: The article starts with an analysis of the major findings of the verdict (Section 2), followed by critical remarks on the FCC's concept of freedom and human rights (Section 3.1), as well as with regard to the Paris Agreement and IPCC budget (Section 3.2). Furthermore, the consequences regarding the protection level and policy measures (Section 3.3), especially with regard to EU climate policy and emissions trading (Section 4), are discussed, which is followed by some short conclusions (Section 5).

## 2. Materials and Methods: Findings of the German Constitutional Verdict

Methodologically, this article provides a legal interpretation of human rights in German constitutional law and of their reading by the FCC. In particular, we will—using the methodology of a legal interpretation of human rights and an analysis of the verdict—elaborate on the weaknesses of the verdict with regard to climate protection. Furthermore, we will show the practical consequences of the ruling. Legal norms are interpreted grammatically, systematically, teleologically, and historically, i.e., according to their literal meaning, their relation to other legal norms, their purpose, and their evolution. Usually, grammatical and systematic interpretation is applied since the other two approaches are prone to several problems. In the Anglo-Saxon legal sphere, case law would also serve as a source of interpretation; thus, implying that such a case (like the FCC case) would be seen as a source of law. This is different to the continental legal sphere we are based in. Therefore, the FCC verdict will also be subject to criticism in the following sections on the basis of an interpretation of human rights as such. A (very long and) detailed analysis of all court rulings on climate change, the 1.5 degree target, and human rights is presented elsewhere [8]. Nota bene: regarding the epistemological background, legal interpretation is—like ethics or practical philosophy—normative science, not empirical science. Law and ethics make statements of ought rather than statements of being. Therefore, legal interpretation does not require experiments, nor the collection of data and facts, i.e., legal

interpretation is not a case study, as a case study empirically describes a process (see in detail [18,19]—also, on the criticism of empiricism in epistemology that suggests since the 17th century that science can only deal with facts, not with norms).

First of all, this section discusses the numerous important points of the German verdict [5] (on the following, see [6,7,10,11,21]). The FCC ruled in favour for more environmental protection in Germany's first successful climate lawsuit: the German legislator has to strengthen emissions reduction targets, which are laid down in the German KSG; measurable interim goals post-2030 also have to be defined; and a sufficient developmental pressure and planning certainty is required so as to shape the transition to a post-fossil stage to be as freedom friendly as possible (similarly, there is also the example of the Irish Supreme Court in the case of Friends of the Irish Environment v. Ireland, [22]). The legislator must raise the level of ambition in climate policy in order to distribute freedom opportunities and the remaining carbon budget for meeting the Paris climate target fairly between generations (see also [23–28]). Consequently, the verdict goes further than the world-famous Dutch Urgenda ruling, which only prohibited lowering a (low) climate protection level once it has been established by the government (see also Leghari v. Federation of Pakistan [29]). The FCC, furthermore, finds that parliament—not government—has to take essential climate policy decisions.

As brought forward in the first constitutional complaint, the verdict justifies climate protection and net-zero-emissions targets with human rights, as well as with the overall German state objective to protect the environment (Article 20a German constitution, i.e., Basic Law) [13]. Human rights demand a comprehensive protection of freedom, as well as the elementary preconditions of freedom [1,2,18,19,30,31]. Thus, the FCC recognises life and health as well as minimum subsistence as human rights (see Article 2(2) Basic Law) in climate protection [18,19,23,32–38]. In this context, the FCC attributes an intertemporal and global, cross-border effect to human rights (paras. 175 and 182 of the verdict; see also [39]). The court does not give a reason for this—although reasons are discussed in the literature and the first constitutional complaint [18,19]. The most important argument— which is possibly relevant for all liberal-democratic constitutions—is that freedom should be effective in any situation where it is threatened—and today, unlike centuries ago, this threat often extends over great distances and periods of time (on intertemporality, at an early stage of the German debate [40], as well as on globality, in the German debate [41]; see an overview of this in [37,42]).

Moreover, the FCC applies its usual approach on the legislature's scope of decision making in climate policy. The role of a constitutional court is to make sure that legislation stays within its balancing limits (see, in more detail, [19,36,43]). These substantive requirements to legislatory balancing are complemented by the procedural requirement that the parliament should take the essential climate policy decisions (as a basis in the German debate, see [44,45] and also [18]; similar with regard to the Urgenda verdict, see [3]). As a basis for the substantive and procedural requirements, the FCC observes that climate protection is about freedom rights as a whole (para. 127 of the verdict)—in two contradictory ways ([25,30,34]): both climate change and climate protection can impede freedom, i.e., a double threat to freedom (on the basis of this in the German debate, see [18,19,46–48]). Therefore, the (substantive and procedural) limits of the legislator's scope of decision making must be examined in both directions. When addressing protection of freedom from negative effects caused by climate change, the court had access to four perspectives that are applicable under liberal-democratic constitutions since the first constitutional complaint: *(1) An argument on the right to the elementary preconditions of freedom to life, health, and minimum subsistence as a protection right obliging the state to protect individuals against their fellow citizens causing climate change; (2) an argument on the same fundamental rights as a defensive right against a state-permitted climate change (by having harmful subsidies, state permissions for coal-fired power plants, cars, etc.); (3) an argument on freedom as a whole in connection with the state objective of environmental protection* [13,49]. In the verdict, the FCC follows the first and the third argument (and completely ignores the second one). However, the court finds that although

climate policy is criticised for being weak, it is still justifiable under constitutional law. This is because the legislature is seen as having a far-reaching discretion in concretising the required environmental protection level under the first and third argument (even though the legislator or the government cannot lower a level of protection once it has been adopted—this is also a parallel to the Urgenda verdict [50,51]).

The FCC recognises that the political agreement on the Paris target is binding under international law (similarly, see also the Administrative court in the French case Notre Affaire à Tous and Others v. France; for a legal interpretation of Article 2(1) of the Paris Agreement, see also [30,52]). According to the court, the Paris target aims to limit the temperature increase to 1.5 degrees Celsius—rather than merely to well below 2 degrees Celsius or even only to 2 degrees Celsius. Yet, the Court still finds that, as a concretisation of Article 20a of the Basic Law, the limit of well below 2 degrees seems to suffice (para. 235 of the verdict; see also [30]). *In contrast to this, the real climate protection obligation is deduced from argument (4), which is based on the protection of freedom as a whole against climate-policy measures, also aiming at balancing freedom over time. The FCC argues that the legislature has not done this so far because it has failed to account for the urgent climate policy after 2030, which would highly endanger freedom given that sooner or later radical, freedom-encroaching climate policy measures become more and more likely.* This crucial distinction between "protection from climate change vs. protection from climate policy" remains unaddressed in the scholarly analyses of the FCC verdict.

With regard to procedural requirements, the FCC not only argues that parliament (not government) must take the essential climate policy decisions, but that politics must also be based on the current state of empirical scientific knowledge such as climate science (similar to the Urgenda verdict, see also [53]). The verdict finds that facts must be carefully examined, even if there are knowledge gaps. Furthermore, knowledge development must be monitored and, if necessary, new political decisions have to be made on this updated knowledge. In the past, the FCC had asserted such fact-finding rules rather vaguely and only occasionally (relatively precise before at [54]; see also [19,55]). Overall, these procedural aspects were also admonished in the constitutional complaints.

The court refuses to clearly accept that a low climate-protection level (and not only "too much climate policy") violates human rights. Still, it makes some very important points on climate protection and human rights (i.e., on the arguments one, three, and four mentioned above). The court accepts an overall constitutional obligation (based on protection rights and the state objective, as well as indirectly on freedom as a whole) to protect the climate—both intertemporally and globally. Furthermore, the FCC applies the precautionary principle to human rights (for further explanations on the precautionary principle in international law, see [1,2,19,27,31,56–59]), and thereby follows the arguments of the first constitutional complaint (paras. 129 et seq. of the verdict; with a critical opinion on this, see [60]; the precautionary principle has also been recognised in the Urgenda ruling, see State v. Urgenda [3], as well as in [19,30,56–58]). This means, not only the present violations of the complainants' human rights, but also cumulative, uncertain, and long-term violations of fundamental rights are relevant. This is convincing because fundamental rights would otherwise be meaningless in the case of imminent, irreversible damage. This is exactly what the FCC recognises (cf. as a basis [7,19]). In the past, the precautionary principle was mostly read as objective law (i.e., obligations of public authorities that nobody can base a lawsuit upon), not of human rights, and it was assigned to norms such as Article 20a of the Basic Law or on Article 191 of the Treaty on the Functioning of the EU alone [61,62]. In contrast, the FCC now argues that human rights are affected even if, as in the case of climate change, many people are affected. The court thus adopts the arguments of the first constitutional complainant (para. 110 of the verdict), and, as a consequence, the discourse on causality in climate protection from the international arena is thus rendered moot. This again seems convincing—because why should the violation of one's human rights be trivial just because other people's rights are also being violated? Similarly, the discussion on the attribution of damage was overruled by the FCC. It found

that there is a legislative duty of the state to act regardless of whether other states also act—climate protection is an international concern in which all states have to participate (paras. 199 et seq. of the verdict; on the debate about causation and attribution in climate litigation see [63]).

Taking the findings presented in the last paragraph into account (which present a rather new concept of freedom and a far-reaching obligation for climate legislation), the FCC verdict is revolutionary. This is also true in view of the substantively far-reaching nature of the verdict, the worldwide perception of the ruling, and the international reputation of the FCC, as well as because the verdict has an indirect legal effect throughout the EU (on this, see Section 4). Apart from an indirect legal effect at the EU level, the FCC's arguments could be, for example, taken up in ongoing international court cases, where similar issues are currently being discussed before the European Court of Human Rights and the International Court of Justice. Nevertheless, considerable criticism could remain, and consequences have to be pointed out more clearly, as we will analyse in the following.

## 3. Results

In the following, we—using the methodology of a legal interpretation of human rights and an analysis of the verdict—elaborate on the weaknesses of the verdict with regard to the concrete climate target, as well as to some aspects of the required new concept of freedom that the FCC did not take into account. Furthermore, we will show the practical consequences of the ruling for climate protection levels and effective policy instruments.

### 3.1. Biased Understanding of Freedom?

Since the legal literature has so far failed to discuss the distinction between "protection from climate change vs. protection from climate policy", one of the central weaknesses of the FCC verdict has been hardly discussed. Contrary to what the court insinuates, the greater danger to freedom and its preconditions is that of politically accepted or favoured climate change—not delaying climate action and then applying a radical climate policy. Climate change may turn food and water supplies into precarious resources in some parts of the world. Natural disasters will become more likely, thus leading to major migratory movements, wars, and civil strife. Moreover, according to conservative estimates, dealing with the consequences of inaction on climate change is expected to be around five times more expensive than action on climate change through ambitious climate policy (besides the IPCC reports, see [33,64,65]; however, these do not consider the most expensive aspect, which is climate wars; see also [66,67]). The court has not addressed these aspects. For example, the court discussed adaptation instead of mitigation as a partly permissible strategy for human rights protection against climate change (para. 181 of the verdict), even though, climate wars (as an example) might be barely controllable by climate-change adaptations. Notwithstanding this, adaptation to climate change—because it is already underway—also remains important.

The FCC verdict arrives at a difficult compromise: On the one hand, there is an obligation to ensure a greater climate protection based on the fundamental rights of intertemporal protection of freedom (see also [27]), including the state objective of environmental protection. On the other hand, the restrictive doctrine of the protective dimension of fundamental rights is maintained without even discussing the criticism provided in the constitutional complaints. Instead, the FCC repeats its own restrictive judgements on the protective dimension of human rights. *If the FCC had considered human rights as defensive(!) rights against the state actively causing climate change—for example with regard to the allocation of emission certificates or via the approval of coal-fired power plants and open-cast mines (the above-mentioned argument (2))—the court could have viewed climate change as the most important human rights problem.* Instead, the FCC considers the (initially delayed and then later) foreseeable radically rapid reduction in emissions as the problem that ultimately triggers the unconstitutionality of climate policy—which provides a conceivably paradoxical derivation of a "human right to climate protection".

This finding seems even more challenging because the arguments against strong protective human rights (i.e., obliging public powers to protect freedom and its preconditions against our fellow citizens) are not convincing under liberal-democratic constitutions: Human rights law considers the defensive and protective dimensions as equal—e.g., in Germany, in Article 1(1) sentence 2 and in Article 2(1) of the Basic Law and in the EU in Articles 1 and 52(1) of the Charter of Fundamental Rights (more detailed on the following [19]; early critical voices on this in the German debate were also [18,36,68,69]). Furthermore, the FCC has not taken into account that claims for protection do not necessarily come with a "claim to some specific legislation". Rather, as in the case of defensive actions, the sole purpose (as argued by the complainants) of a claim can be set to an outer boundary through a judicial finding—i.e., "not like this" instead of "do exactly that". When discussing the separation of powers, the protection of human rights against the state (defensive dimension) and the protection of fundamental rights against fellow citizens by the state (protective dimension) may therefore not differ at all. The role of a constitutional court is the same, i.e., to not prescribe concrete policy instruments, but to define the limits of legislative leeway and to demand compliance with these limits.

This argumentative approach creates a strange imbalance in the FCC's verdict. On the one hand, the court emphasises Article 20a of the Basic Law and protection rights against climate change under the heading of protection of freedom as a whole. On the other hand, the court finds that the present, less ambitious German Klimaschutzgesetz is still compatible with protection rights and with Article 20a in Basic Law. However, the finding that a state objective (not protection rights!) has little practical impact and is largely left to legislative concretisation is quite consistent. State objectives have fewer structures than human rights. In case of human rights, the balancing limits (in substantive and procedural terms) of the legislature's action scope can be derived from those very rights, and these limits may be called, for example, the four levels of proportionality or be subdivided more precisely and named as rules or limits to balancing. Thus, for example, if rights to the elementary preconditions of freedom (regardless of whether they are understood as defensive or protection rights) collide with the economic freedom of occupation, action, and ownership of entrepreneurs and consumers, no one may be deprived of more freedom than is necessary to increase the freedom of others (the so-called principle of appropriateness and necessity). Likewise, no balancing results are allowed that could undermine the physical preconditions of future democratic balancing processes (as a sub-aspect of the rule of appropriateness). In the present case, the latter could have plausibly justified a verdict to demand more ambitious climate policy, i.e., a justified demand due to lacking climate protection rather than threatening rushed climate policy.

In other words—in the case of Article 20a of the Basic Law, it remains open as to how a concrete protection standard may arise from the general requirement that the state protects the basis of life. In the verdict, the FCC argues that Article 20a of the Basic Law (just as human rights) is a legal principle (see also [70–73]). Therefore, questions would have to be asked as to (a) whether the scope of protection is impaired and (b) whether the impairment is justified by other legal interests within the framework of balancing limits. Since there is no legal standard for (b) in the German constitution (in contrast to human rights that provide balancing rules by interpreting in particular the concept of freedom and other hints in the constitutional wording), the result of the FCC is unsurprising: ultimately, there is no violation despite there being very unambitious climate-protection policies. The question is also not answered by the FCC's reference to adapting natural scientific findings (cf. [10,11,49]); this is because facts alone do not provide normative criteria. Rather, facts provide subsumption material (see also [50]). And even a normative duty (which is equally obvious for human rights and a state objective) to carefully examine the natural data alone does not provide any action guidance as long as it is unclear which normative standard must be used to determine if the legislator has protected the environment sufficiently. Without such a standard, the FCC's view that the legislature can more or less choose the protection standard seems unsurprising with regard to the balance of powers. Still, the

court then discusses a potential solution that was also outlined by the complainants. An interpretation referring to Article 2(1) of the Paris Agreement may be used to determine the protection standard ([74]). However, this approach would have been even better suited to contouring the human rights of (either of defence or protection of) life, health, and subsistence, which is where balancing limits would have been available. These balancing limits are mentioned by the Court only with regard to the protection of freedom against postponed and then radical climate policy (for example in para. 246 of the verdict).

Furthermore, the FCC could have taken a closer look at the Paris target, as also submitted by the complainants, and arrive at different legal interpretations of this norm and of fundamental rights (and/or the state objective of environmental protection).

### 3.2. Paris Temperature Target and IPCC Budget?

The FCC recognises the binding nature (at least under international law) of the political agreement on the Paris target as a global climate target, i.e., making efforts to limit global warming to 1.5 degrees. The FCC understands Article 2 PA as a concretisation of the constitutional level of climate protection and as a binding for the legislature itself (paras. 235 and 242 of the verdict; on the Paris target in detail (which will not be repeated in the present article; see [1,2,30]). Furthermore, the court points out that Article 2(1) of the Paris Agreement does not only refer to keeping temperature below "two degrees" (unlike most of the literature; cf., for example without justification, see [75]). Instead, states must attempt to comply with a limit of a 1.5 degree Celsius increase.

To underpin the 1.5-degree Celsius limit, the FCC has adopted the approach of the IPCC [76] and the German government's Council of Environmental Experts (SRU) (cf. [77]). Both institutions calculated a greenhouse gas budget to stay within the 1.5-degree Celsius limit. This again seems principally convincing, but the FCC has ignored the weaknesses of the IPCC budget. These weaknesses result from the IPCC being a consensus body that works with optimistic assumptions (e.g., on climate sensitivity and tipping points; on all empirical criticisms see [1,30,78–80]). The FCC refers generically to the fact that the greenhouse gas budget could be calculated too high or too low—despite the verdict making a strong case for policymakers to carefully consider scientific evidence (rather determined insofar the ruling FCC in [54]; fundamentally [18,19,55]). The FCC has also ignored the legal criticism of the IPCC budget, which is intended to concretise a legal norm, i.e., Article 2(1) of the Paris Agreement (on this and the following with further references, see [1,2,30,81]; the latter means that the IPCC's figures of [82] do not change the essence of the criticism; the following points are passed over at [75]). Article 2(1) of the Paris Agreement is legally binding [1,30,83–85], as the court itself presupposes (that this is true can be derived from Article 3 and 4(1) of the Paris Agreement [1,30]). However, against this background, it is insufficient to aim at a 1.5 degree Celsius limit only with a 67 percent probability of being achieved. Nota bene: it is irrelevant for this legal obligation that it is currently becoming increasingly difficult to meet the target due to the unwillingness of most states.

Revising the compliance probability to 83 percent, as conducted in the sixth assessment report of the IPCC in 2022 (AR6), would reduce the budget to 300 GtCO2 globally as of 01.01.2020 (for a closer review on the figures of [76,82], see [30]). On a per capita basis, the remaining German budget is 3 GtCO2 ([86]; see also [87,88]). Given the large annual emissions of Germany, this budget will be used up by 2023–not in ten years, as the FCC presupposes. The budget decreases further if a higher probability is adopted or if other problems of the budget are addressed, such as the base year or the unequal distribution of the budget towards countries of the Global South. According to its wording, Article 2(1) of the Paris Agreement refers to the comparison with the pre-industrial level. For this purpose, however, a year in the second half of the 19th century cannot be chosen as the base year—as was chosen by the IPCC—because industrialisation started gradually from around 1750. The fact that only estimates and no measured data are available for the first hundred years of industrialisation does not rule out this argument (see also [89]). The IPCC budget

decreases even further if a temporary overshoot is excluded and more cautious empirical assumptions are taken, especially with regard to tipping points or climate sensitivity. This is because new scientific findings—which must always be carefully taken into account by the legislator—show, with increasing certainty, that climate change is progressing even faster and will deliver more dramatic consequences, including a collapse of the Gulf Stream, which is vital for Europe (see, currently, the study of [90]).

Against this backdrop, it is particularly surprising that the FCC erroneously stated that the complainants never criticised the IPCC budget (they did, especially the first constitutional complaint) (para. 223 of the verdict). Another budget question concerns whether or not each human being on Earth should be allocated an equal share of the remaining emissions budget ("one human, one emission right"). On the one hand, distributional issues in liberal democracies do not usually require a strict equal distribution. On the other hand, the distribution of climate emissions could be a case to at least aim at an equal distribution over longer periods of time. The FCC correctly points out that Article 2(2) and 4(4) of the Paris Agreement tend to argue for an unequal distribution at the expense of industrialised states (para. 225 of the verdict). These standards are based on capability and historical causation. If these aspects were taken into account, the German or European budget would have already been used up—or there would be an implication of an obligation to contribute massively to emission reductions outside of Europe to compensate for the emissions outside the budget (for this topic in more detail, see [1,19,30,43,91]).

Overall, if we take the FCC's parameters seriously—i.e., the duty to carefully ascertain the facts, the carbon budget concept, the binding nature of the 1.5-degree Celsius target under international law at the very least (as well as the associated interpretation of the Basic Law), and the necessity for an unequal distribution of the budget towards the Global South—we arrive at a very small carbon budget even if emission rights are bought from other countries.

### 3.3. Protection Level and Policy Measures

The far-reaching effect of the FCC ruling addresses public authority as a whole, i.e., not only the legislature, but also the administration and judiciary sectors (and their interpretation of administrative law, civil law, etc.). They all have to strive for climate neutrality and the intertemporal protection of freedom (see also [25,27,32,40]). This includes federal legislators, federal governments, regional legislators, subordinate authorities, local governments, and courts, as well as—indirectly—the EU level (see next chapter).

Under climate constitutional law, the double threat to freedom implies an obligation for an ambitious budget. This obligation is linked to an obligation to create planning perspective and certainty in order to carefully determine the natural scientific basis and to respect the requirement of parliamentary approval. In doing so, the legislature must carefully review facts repeatedly. It must also take into account that the criticism of the IPCC budget and the arguments for a globally unequal distribution of the remaining budget (along the lines of capability and historical causation) demand a significantly smaller carbon budget than is usually presupposed (see above). These findings result from a legal interpretation of core terms of liberal-democratic constitutions, and they are therefore highly relevant for other jurisdictions nationally and transnationally. In 2021, Germany tightened its climate goal directly in response to the verdict from −55% to −65% greenhouse gas emissions by 2030 compared to 1990. However, when measured against the aforementioned obligation to carefully gather facts and against what was said above about the 1.5 degree limit, this goal is still not ambitious enough.

Overall, these commitments imply a comprehensive fossil-phasing-out strategy in all sectors, i.e., in industry, energy, transport, buildings, and agriculture. Livestock farming has to also be reduced substantially. These measures need to be supplemented by safe measures for negative emissions such as in forestry and peatland management to compensate for residual emissions from industry and agriculture (on negative emissions, see [2,92,93]). However, while the FCC mistakenly emphasises the emerging threats of

future climate policy, it forgets that avoiding climate warming promises to be economically far more favourable than climate catastrophe, thus it cannot be portrayed as a mere burden (traditionally and unchanged to this, see [64]; for a more careful account, see [94]).

Even if the FCC only addresses targets and ambition levels, the verdict also has implications for policy measures because climate protection strongly depends on policy measures. Still, when taking balancing leeway and separation of powers into account, it will remain difficult to sue parliament for a specific policy measure in the constitutional court. However, the court can assess the extent to which policy measures adopted by legislative bodies are within the limits set by the established protection level, as well as by the necessity for planning perspectives and the obligation to carefully ascertain the facts (including natural sciences and insights in the effectiveness of different policy measures). Staying within the temperature limits of the Paris Agreement requires phasing out fossil fuels in all sectors (electricity, heat, mobility, agricultural sector, cement, plastics, etc.), as well as strongly reducing livestock farming and having compensation for residual emissions, as has been seen. Furthermore, the legal competencies and the facts regarding the effectiveness of different policy levels could imply that a national government and parliament have to push for effective solutions, especially at the EU level (see next chapter).

The character of the general problem and of liberal-democratic legal interpretations imply further proceedings at the FCC—at the protection level and on the framework conditions for effective policy measures. The same applies to the European Court of Human Rights and the International Court of Justice in ongoing proceedings in 2023–and constitutional courts in other countries (cf. [95]; for more cases in other countries, see also [96,97]). And even if the European Court of Justice–the constitutional court of the EU–continues to pursue its narrow understanding of the formal requirements of actions for annulment under Article 263(4) in the Treaty on the Functioning of the EU, it will most likely be confronted with the question of whether it will give EU primary law an interpretation that is similar to the one presented here. This would be plausible since EU primary law also constitutes a liberal-democratic constitutional order. Likewise, the UN Human Rights Council adopted a (non-binding) resolution in October 2021, which recognises a right to a clean environment [98]. However, given the unclear content of such a right on the one hand, and the clearly identifiable content of traditionally recognised human rights with regard to climate on the other hand, such an initiative appears less promising.

## 4. Discussion

The FCC emphasised that Germany must push for climate protection internationally and must not claim that others do not push. The reason for this—beyond the fact that unilateral inaction makes inaction by other states more likely—is only discussed rudimentarily, but it is clear nevertheless (in more detail, see e.g., [19,20]). Firstly, climate warming cannot be solved in Germany alone: global warming is a global issue [99]. Secondly, purely national climate policy threatens to trigger sectoral and spatial shifting effects (the well-known keyword of carbon leakage refers to the spatial component), which would be ecologically counterproductive and could undermine the acceptance of climate protection as a whole due to competitive disadvantages (an EU climate policy could also trigger shifting effects, at least outside Europe, but these can be avoided by border adjustments because of the EU's customs competence; see [19]). Thirdly, purely national climate policy is already legally impossible due to the legal competences of the EU: many emissions are fully regulated under EU law, for example within the framework of the EU emissions trading scheme (once again, these aspects and the following arguments seem to also apply for other liberal-democratic jurisdictions). Therefore, the first constitutional complaint explicitly requested the FCC to declare that Germany had not sufficiently pushed for climate protection at the EU level.

However, the FCC does not explicitly mention that most emissions are not regulated by German legislation alone, but by EU legislation. However, in view of the obligation to transnational climate protection, to observe facts carefully (including the question of the most effective policy level for climate protection) and the described impossibility of tackling

climate protection while neglecting the EU level, the following applies (even without an explicit statement by the FCC): Public authorities such as the German Federal Government must also try to enforce their domestic (in this case climate) constitutional requirements via legislative procedures at the EU level. A nation state is obliged to push for more effective EU climate protection.

The ambitious protection level and the findings on the promising (EU) policy level, as well as on the effectiveness of various policy instruments, point towards focusing on optimising EU climate policy and, especially, the EU emissions trading scheme. This can be seen as an implication of the FCC verdict, at least, if the factual situation almost inevitably demands this instrument. Recent research findings have shown that a further expanded and restructured EU emissions trading system offers the best guarantee through which to comply with the required protection level. In addition, a reformed EU emissions trading system avoids governance problems if it is designed differently and meets the requirement of a freedom-preserving transition to post-fossilisation (for a more in detail account on the following aspects, see [19], as well as [100–104]). This is achieved by *setting ambitious caps and by addressing easily detectable governance units (such as fossil fuels or animal products at the level of slaughterhouses and dairies) on a sectoral and geographically broad scale (i.e., at the EU level plus with climate clubs in association with other countries plus border adjustments).* Reformed along these lines, emissions trading may avoid governance problems such as enforcement, rebound, shifting effects, and problems of depicting emissions more effectively than any other governance instrument (for an account on behavioural research as a basis for identifying the governance problems mentioned above, see [19,100,101]). In contrast, regulatory law focuses on individual products, activities, or facilities, and it is thus exposed to rebound and (sectoral and spatial) shifting effects, as well enforcement problems that can undermine the desired reduced ecological footprint or—in the worst case—can turn it into its opposite. Furthermore, quantity governance may encourage more consistency, resource efficiency, and frugality as sustainability strategies: if the cap is not achievable purely technologically, addressees will inevitably switch to frugality (for a more detailed account, see [19,100,101]).

Furthermore [19,100–105], cap-and-trade approaches may comprehensively address the motivational situation of citizens. This not only includes monetary self-benefit, but also conceptions of normality—such as "going by car and having a big piece of meat every day is normal"—as well as emotional factors such as convenience and denial. In addition, quantity governance is particularly compatible with the basic principles of liberal democracies because it maintains the greatest possible degree in the freedom of consumers and enterprises, while at the same time effectively defending the physical preconditions of freedom against the double threat to freedom. Furthermore, quantity governance may well be combined with—national or transnational—social distributive measures (as compensation for the distributional effects of climate change on the one hand and climate policy on the other hand). The fixed cap of an emissions trading scheme prevents redistribution from undermining the ecological effects of the system—something that cannot be avoided by environmental fees with revenue redistribution [102–104].

All these aspects are rarely taken into account. Instead, the focus is usually on cap-and-trade systems' promise to achieve a sustainability goal very efficiently, i.e., "at particularly low cost". Given that the findings quoted above are convincing, the FCC statement in favour of factual accuracy speaks in favour of a stronger regulatory focus on cap-and-trade schemes. Choosing the central drivers of diverse environmental problems such as fossil fuels, animal products, or pesticides as the governance units of cap-and-trade systems, may lead to an integrated solution for most environmental problems [92,93,100,101]. However, this statement only applies if the quantity governance is designed as described (i.e., by setting ambitious caps and addressing the easily detectable governance units on a sectorally and geographically broad scale).

The EU emissions trading system to date does not meet these criteria [102]—even though the reform in 2022 would hardly have been possible without court rulings such as

the one from the FCC. For example, the cap does not correspond to the 1.5-degree Celsius limit, and the system does not cover all sectors. The scheme suffers from loopholes and many old certificates; livestock farming is not covered at all. Lastly, sufficient protection against emissions shifting outside the EU is also missing. In order to achieve the legally binding greenhouse gas neutrality as established in the European Green Deal (which is currently set to be achieved by 2050), the EU presented a series of legislative amendments under the heading "Fit for 55" (see on the following details [105]). The aim is, inter alia, to incorporate fossil fuels almost entirely into the emissions trading system, as well as to tighten the cap and introduce a border adjustment. A social compensation mechanism would supplement this system. These proposals are roughly in line with the findings on the effective emissions trading discussed above. Effective EU climate policy—in line with the FCC's requirements—presupposes that a kind of global climate club is formed simultaneously with many other states that take similar measures. Furthermore, carbon border adjustments have to be introduced against those states that do not participate in the climate club to prevent ecologically problematic and economically disadvantageous shifting effects. If EU emissions trading is to have its potential effect—in line with the FCC's stipulation of a fair intertemporal balance of freedoms and the 1.5-degree limit being binding under international law—Germany would arguably have to urge for improvements. Improvements include an even stricter cap [8,30,106–109], one that is based on a significantly smaller budget. In addition, cancelling most of the old certificates that the states used to give away to companies and that still relativise the effectiveness of the cap would need to be conducted (for a more detailed account, see [91]). Furthermore, an emissions trading approach is needed for animal products; one that is designed in such a way that remaining agricultural residual emissions can be compensated for by measures such as improved forestry or peatland management (for a more detailed analysis, see [100]). However, land use as a whole—i.e., the entirety of agriculture, forests, and peatlands—cannot be covered by a separate emissions trading scheme due to its large heterogeneity; moreover, the problem of depicting emissions opposes peatland certificates or humus certificates [92,93,110–112]. Instead, an emissions trading system that addresses the drivers of peatland and forest destruction (especially fossil fuels and animal husbandry) combined with subsidy and regulatory law may seem more promising (this and the insufficient LULUCF framework have been discussed in detail elsewhere) [92,93,110–112].

## 5. Conclusions

The German Constitutional Court's climate verdict calls for a fair intertemporal balance of freedom opportunities. By demanding more climate protection and not just prohibiting a lowering of an already low climate protection standard, the court's decision clearly goes beyond the Urgenda ruling. Fundamental rights are accepted as intertemporal and globally applicable. The Court understands these rights in light of the precautionary principle and freed from the misleading climate causality debate. Thus far, however, the verdict is rarely understood comprehensively. Overall, the court recognises that, at the centre of effective climate policy, there is the legally binding 1.5-degree Celsius limit from Article 2 of the Paris Agreement. However, the court omits to highlight that Article 2 implies a radically smaller remaining carbon budget. In addition, the court fails to address the most important human rights violations. It misleadingly categorised climate policy as a greater threat to freedom than climate change itself. Furthermore, little attention has so far been paid to the fact that the ruling indirectly imposes an obligation for more EU climate protection, especially since most emissions are regulated at the EU level. The effectiveness of the reformed EU emissions trading system will play a key role, and its reform should go well beyond existing EU proposals.

These findings have consequences in Germany, the EU, and beyond–due to indirect legal effects and because the arguments on freedom and climate protection are transferable to other liberal democratic constitutions due to their general character. The discussion on freedom, climate, ambition levels, and policy instruments of climate protection is of an

incredibly great importance; one that is far beyond the circle of lawyers alone. It will be interesting to see whether other constitutional courts and the European Court of Human Rights, as well as the International Court of Justice, will move further in this direction in some of the ongoing trials in 2023.

**Author Contributions:** F.E. developed the underlying human rights and climate law perspective since 2000; conceptualised and drafted the article; and derived the critical perspectives on the FCC ruling. M.B. contributed to the analysis and edited the paper. All authors have read and agreed to the published version of the manuscript.

**Funding:** This research was partly funded by the Environmental Federal Agency of the German Federal Government, as well as by the earlier fundings that supported the legal opinions of Felix Ekardt on climate litigation since 2010 by the Solar Radiation Promotion Association.

**Institutional Review Board Statement:** Not applicable.

**Informed Consent Statement:** Not applicable.

**Data Availability Statement:** Not applicable.

**Acknowledgments:** We would like to thank Katharine Heyl for proofreading.

**Conflicts of Interest:** The authors declare no conflict of interest.

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
