# Peer review of "The German Climate Verdict, Human Rights, Paris Target, and EU Climate Law"

_sustainability, doi:10.3390/su151712993_

Round 1

Reviewer 1 Report

I found the idea of manuscript super interesting. The German climate Verdict gives a lot of food for thoughts and could interpreted and used as an example for other claims, movements, legislations, etc. However, the text is “very heavy, which makes it difficult to follow. I suggest to simplify the text partly. Include scheme of the analysis, table to present Verdict’s pros and cons, etc. The less usage of ‘law terms’ does not make the manuscript worse, oppositely.

It could be useful to include principal scheme of the analysis which could navigate the readers better through the text. Also, I found a lot of criticism and paradoxes of the Verdict presented. However, I believe that sometimes human rights and climate change could be supportive. Could you provide some other opinions how the Verdict helps on climate change perspective? I do not agree that adaptation part is considered as NOT important factor for human rights and climate change.

It is very hard to distinguish what is new and what is a replica of previous works. I suggest not to use so many self-references, but try to highlight the fresh ideas. I do not found methodology part in this manuscript. It is partly belonging to ‘Introduction’ and partly to ‘Discussion’. Therefore, ‘Results’ section and subsections become not clear. If you want to include the Verdict you can do it as supplement material; presenting main outcomes is not a methodology of the manuscript.

It could be helpful to clearly formulate the main aim of the article, the roadmap of your ideas which will help to follow the main narrative. I was lost in the text and main ideas while reading. And what is new about the Verdict, because you in this topic and research more than 10 years already?

The Verdict was reached more than 2 years ago. You wrote that is very important. So please, provide new examples of legislations, activities (not only resonance courts), which proves the importance. Also, I found the analysis of the Verdict a little bit outdated. How long can we discus about PA 1.5 °C; Five or ten years more? So what if we will prove that the whole world and Germany does not achieve the target. It is already useful to talk only about 2 °C limit. Please, update the manuscript according recent situation.

Author Response

Reviewer 1

I found the idea of manuscript super interesting. The German climate Verdict gives a lot of food for thoughts and could interpreted and used as an example for other claims, movements, legislations, etc. However, the text is very heavy, which makes it difficult to follow. I suggest to simplify the text partly. Include scheme of the analysis, table to present Verdict’s pros and cons, etc. The less usage of ‘law terms’ does not make the manuscript worse, oppositely.

It could be useful to include principal scheme of the analysis which could navigate the readers better through the text. Also, I found a lot of criticism and paradoxes of the Verdict presented. However, I believe that sometimes human rights and climate change could be supportive. Could you provide some other opinions how the Verdict helps on climate change perspective? I do not agree that adaptation part is considered as NOT important factor for human rights and climate change.

It is very hard to distinguish what is new and what is a replica of previous works. I suggest not to use so many self-references, but try to highlight the fresh ideas. I do not found methodology part in this manuscript. It is partly belonging to ‘Introduction’ and partly to ‘Discussion’. Therefore, ‘Results’ section and subsections become not clear. If you want to include the Verdict you can do it as supplement material; presenting main outcomes is not a methodology of the manuscript.

It could be helpful to clearly formulate the main aim of the article, the roadmap of your ideas which will help to follow the main narrative. I was lost in the text and main ideas while reading. And what is new about the Verdict, because you in this topic and research more than 10 years already?

The Verdict was reached more than 2 years ago. You wrote that is very important. So please, provide new examples of legislations, activities (not only resonance courts), which proves the importance. Also, I found the analysis of the Verdict a little bit outdated. How long can we discuss about PA 1.5 °C; Five or ten years more? So what if we will prove that the whole world and Germany does not achieve the target. It is already useful to talk only about 2 °C limit. Please, update the manuscript according recent situation.

Dear Reviewer,

Thank you for taking the time to review our manuscript.

We added and revised the following aspects: We explained the analysis scheme and core ideas in more detail in the first two chapters (even though the methodology has already been mentioned so far right at the beginning of chapter 2). Furthermore, we have highlighted the far-reaching and helpful dimensions about the verdict more directly. We have also changed the headings, added references to new legislation and underlined the role of adaptation more strongly. We have discussed 1.5, well below 2 degrees or exactly 2 degrees in previous articles; we have now referred to this. As far as the category ‘materials and methods’ is concerned, we have used this in the same way as we have done for many MDPI articles in recent years. However, we have thoroughly revised the language there and in the entire article.

Kind regards

Reviewer 2 Report

While this is a potentially very interesting paper at the moment it suffers from a few serious problems:

1. This appears to be written as if for a law journal and an audience of legal scholars. I do not believe that this is the primary readership of this journal. There is a lot of typical dense citing of law and legal precedent, but most of this obscures you key argument: that the FCC's decision in this German case offers profound challenges for the protection of human rights in the face of climate change. I really think you need to identify the key strands of your argument and focus on identifying and analyzing those without getting bogged down in the minutia of the legal details. What is really important about this issue? And what should a non-legal audience focus upon and learn from? How is this relevant to non-German audiences? There is some discussion of this last but again it is lost in the detail.

2. The writing is profoundly dense, cumbersome and not well translated in to colloquial English. Again a hyper focus on the legal context leads to some profoundly obscure writing which will cause most readers to drop off after the Introduction. Again, this is not a law journal, what are the critical elements that need to be presented to the reader?

3. There is a tonal matter, some phrasing that is judgmental and snotty. It is clear that the authors fee strongly about what happened in this judgment but make the case for your argument without judgmental language.

I think the issue is an important one, but again write for the audience of this journal.

This really needs to be edited by someone who is fluent in colloquial English, as it is the translation is rough.

Author Response

Reviewer 2

While this is a potentially very interesting paper at the moment it suffers from a few serious problems:

  1. This appears to be written as if for a law journal and an audience of legal scholars. I do not believe that this is the primary readership of this journal. There is a lot of typical dense citing of law and legal precedent, but most of this obscures you key argument: that the FCC's decision in this German case offers profound challenges for the protection of human rights in the face of climate change. I really think you need to identify the key strands of your argument and focus on identifying and analyzing those without getting bogged down in the minutia of the legal details. What is really important about this issue? And what should a non-legal audience focus upon and learn from? How is this relevant to non-German audiences? There is some discussion of this last but again it is lost in the detail.
  2. The writing is profoundly dense, cumbersome and not well translated in to colloquial English. Again a hyper focus on the legal context leads to some profoundly obscure writing which will cause most readers to drop off after the Introduction. Again, this is not a law journal, what are the critical elements that need to be presented to the reader?
  3. There is a tonal matter, some phrasing that is judgmental and snotty. It is clear that the authors feel strongly about what happened in this judgment but make the case for your argument without judgmental language.

I think the issue is an important one, but again write for the audience of this journal.

Dear Reviewer,

Thank you for taking the time to review our manuscript.

We tried to significantly improve the readability and linguistic quality of the text. Furthermore, we have highlighted more strongly why the subject of our contribution is important for non-legal readers. Irrespective of this, we would like to emphasise that we have published over ten articles at MDPI since 2018, all of which deal with law and governance issues, without this ever being a problem. Moreover, the present text is intended for the Special Issue "Transformation to Sustainability" (co-edited by Felix Ekardt), which is exclusively dedicated to human science issues - i.e., including legal issues. In general, there frequently seem to be different understandings as to how a text is to be assigned - in an academic world that is still very "disciplinary". For example, a purely legal readership would possibly expect not less, but even more details in our text.

Kind regards

Reviewer 3 Report

The paper compares examines a recent German Constitutional Court’s climate verdict calling for a fair intertemporal balance of freedom opportunities. In particular by demanding more climate protection and not just prohibiting a lowering of an already low climate protection standard, the court’s decision clearly goes beyond the “business as usual” as we know it. However much for the profound implications of this court’s ruling are not widely discussed or even understood. So aim of the paper is to shed some light into this content.

I believe that the paper is very well written and could be considered for publication into the journal (Sustainability). However it actually looks at some aspects not really dealing with engineering or technical but more jurisdictional and political: while this is very useful for understanding and guiding policy-makers, it is much less engineers and technical people. So I think it might not be considered appropriate for this specific MDPI journal.

Very interesting is what pointed out by the FCC regarding human rights and their “extention” beyond cross - geographical borders (“In this context, the FCC attributes human rights to have an intertemporal and global, cross-border effect”) . While plenty of references are provided, this point may spark discussion and criticism by other European and non-European nations. Using as reference the international law, I have doubts this idea would be consequential.

Another controversial sentence is “The FCC verdict is revolutionary – _even from a global perspective.” as there is no way that a German sentence could be applied in other countries without violating their respective sovereignty and integrity (legally speaking) , nor can be enforced.

I believe that the problem is very described requiring essentially no revisions before it can be published, but I wonder if a journal with much more political aim would be more appropriate for the topics discussed.

Author Response

Reviewer 3

The paper compares examines a recent German Constitutional Court’s climate verdict calling for a fair intertemporal balance of freedom opportunities. In particular by demanding more climate protection and not just prohibiting a lowering of an already low climate protection standard, the court’s decision clearly goes beyond the “business as usual” as we know it. However much for the profound implications of this court’s ruling are not widely discussed or even understood. So aim of the paper is to shed some light into this content. I believe that the paper is very well written and could be considered for publication into the journal (Sustainability). However it actually looks at some aspects not really dealing with engineering or technical but more jurisdictional and political: while this is very useful for understanding and guiding policy-makers, it is much less engineers and technical people. So I think it might not be considered appropriate for this specific MDPI journal.

Very interesting is what pointed out by the FCC regarding human rights and their “extention” beyond cross-geographical borders (“In this context, the FCC attributes human rights to have an intertemporal and global, cross-border effect”) . While plenty of references are provided, this point may spark discussion and criticism by other European and non-European nations. Using as reference the international law, I have doubts this idea would be consequential.

Another controversial sentence is “The FCC verdict is revolutionary – _even from a global perspective.” as there is no way that a German sentence could be applied in other countries without violating their respective sovereignty and integrity (legally speaking) , nor can be enforced.

I believe that the problem is very described requiring essentially no revisions before it can be published, but I wonder if a journal with much more political aim would be more appropriate for the topics discussed.

Dear Reviewer,

Thank you for taking the time to review our manuscript.

We have published over ten articles at MDPI since 2018, all of which deal with law and governance issues, without this ever being a problem. Moreover, the present text is intended for the Special Issue "Transformation to Sustainability" (co-edited by Felix Ekardt), which is exclusively dedicated to human science issues - i.e., including legal issues. Nevertheless, we have now emphasized more strongly the importance for a broad readership in the text. We have also added the ways in which a court decision (not only, but especially within the EU) influences other countries. The discussion on the extent to which fundamental freedom has an intertemporal and cross-border effect has been conducted in detail elsewhere by us (and others). We have nevertheless added something to this as well.

Kind regards

Reviewer 4 Report

The manuscript entitled “The German Climate Verdict, Human Rights, Paris Target and EU Climate Law” and submitted to the Editorial office of Sustainability presents a discussion around the important issues related to climate policy, analyzing German Federal Constitutional Court statements, justifications, weaknesses and political consequences of its ruling. The current version of the article contains serious flaws and is not suitable for publication. Authors should study properly the Instruction for Authors and revise the manuscript accordingly.

1.      First of all the article is not well-structured. Please note, that some sections are mandatory, and their titles cannot be changed (e.g. Materials and Methods).

2.      The Authors should underline clear the objective and research novelty of the paper.

3.      Abstract should contain: 1) Background; 2) Methods; 3) Results 4) Conclusion. The abstract should be an objective representation of the article.

4.      Materials and Methods should be described with sufficient detail to allow others to replicate and build on published results.

5.      The presentation of the paper is not good, section Results doesn’t provide clear results obtained by the authors. The manuscript is submitted as Article. Please note, that articles are original research manuscripts. The work should report scientifically sound experiments and provide a substantial amount of new information. The current version of the manuscript doesn’t correspond to this type of submission.

6.      The major concern is that a discussion raised around the legislative issues is more suitable for the journal specialized in politics and law.

Author Response

Reviewer 4

The manuscript entitled “The German Climate Verdict, Human Rights, Paris Target and EU Climate Law” and submitted to the Editorial office of Sustainability presents a discussion around the important issues related to climate policy, analyzing German Federal Constitutional Court statements, justifications, weaknesses and political consequences of its ruling. The current version of the article contains serious flaws and is not suitable for publication. Authors should study properly the Instruction for Authors and revise the manuscript accordingly.

  1. First of all the article is not well-structured. Please note, that some sections are mandatory, and their titles cannot be changed (e.g. Materials and Methods).
  2. The Authors should underline clear the objective and research novelty of the paper.
  3. Abstract should contain: 1) Background; 2) Methods; 3) Results 4) Conclusion. The abstract should be an objective representation of the article.
  4. Materials and Methods should be described with sufficient detail to allow others to replicate and build on published results.
  5. The presentation of the paper is not good, section Results doesn’t provide clear results obtained by the authors. The manuscript is submitted as Article. Please note, that articles are original research manuscripts. The work should report scientifically sound experiments and provide a substantial amount of new information. The current version of the manuscript doesn’t correspond to this type of submission.
  6. The major concern is that a discussion raised around the legislative issues is more suitable for the journal specialized in politics and law.

Dear Reviewer,

Thank you for taking the time to review our manuscript.

As requested, we have restructured the abstract and emphasised the novelty of the findings. Furthermore, we have published over ten articles at MDPI since 2018, all of which deal with law and governance issues, without this ever being a problem. As is customary in these subjects, our articles have never contained experiments (even though MDPI does not offer a separate article format such as "perspective"). Small variations in the headings have always been tolerated in the past; this is also unavoidable because the MDPI scheme is largely oriented towards natural scientific research. Nevertheless, we have now adapted the headings more closely to the MDPI scheme.

The present text is intended for the Special Issue "Transformation to Sustainability" (co-edited by Felix Ekardt), which is exclusively dedicated to human science issues - i.e., including legal issues.

Kind regards

Round 2

Reviewer 1 Report

I am very glad with improvements and adjustments of the previous version of the manuscript. Now, it is easier to follow and highlight the most important findings The text language become easier to follow for not law professionals. I encourage to concentrate your efforts not only academic discourse of Climate Litigation, but try to reach bigger audience (not only scientists) with these messages.

Author Response

Dear editors, dear reviewers,

Thanks a lot for taking the time to improve our manuscript.

From the reviewers' feedback, we currently see no further indications for further changes to the text:

  • Reviewer 3 has made no further comments.
  • Reviewer 1 was very pleased with our revision and did not ask for any further changes (but just suggested in general that we go public with our findings beyond this article). We thank you very much for this feedback and will take it into account in our further work.
  • As regards Reviewer 2: We did not write (as the assistant editor quotes him/her in an e-mail) that we find Reviewer 2's comments irrelevant. Please see our answer in review round one - we have indeed revised the text in response to Reviewer 2's helpful comments. He or she does not mention any new points of criticism now. Therefore, no concrete need for revision arises from Reviewer 2's comments.
  • Reviewer 4 writes: “Unfortunately the revision done is not sufficient to make the article corresponding to the journal rules and attractive to the broader audience. In my opinion the paper is more suitable for the journal specialized in politics and law.” Thank you for your feedback, but we respectfully disagree with this point of view. We first refer again to our answers in review round one, from which the criticism of Reviewer 4 in the first round and our responses and - major - revisions on our text can be seen. The new review statement does not express any detailed criticism now, but finds that Sustainability/ MDPI is not a journal for political and legal questions of sustainability. Again, this does not lead to any specific need for revision of our article. I and my co-authors have published over ten articles in MDPI journals such as Sustainability, Land or Environments on questions of law and policy of sustainability since 2018. We don't see why this should no longer be possible. In the end, this seems to be a question to the editors how to deal with this. If the editors seriously agrees with Reviewer 4, we would unfortunately have to cancel the entire Special Issue. Because a human sciences Special Issue - which inevitably also deals with law and politics - would then no longer make sense.

Best regards,

Felix Ekardt

Reviewer 2 Report

n/a

n/a

Author Response

(The authors gave the same response as above.)

Reviewer 4 Report

Unfortunately the revision done is not sufficient to make the article corresponding to the journal rules and attractive to the broader audience. In my opinion the paper is more suitable for the journal specialized in politics and law. 

Author Response

(The authors gave the same response as above.)
